

# Mode-specific effects of concentric and eccentric isokinetic training of the hamstring muscle at slow angular velocity on the functional hamstrings-to-quadriceps ratio-a randomized trial

Kushla Nand Sharma[1], Nishat Quddus[2], Unaise Abdul Hameed[3], Sohrab Ahmad Khan[2], Anita Kumari[4], Ahmad H. Alghadir[5] and Masood Khan[5]

[1] Kataria Healthcare, Sadhya Physiotherapy Clinic, New Delhi, India
[2] Department of Rehabilitation Sciences, Jamia Hamdard University, New Delhi, Delhi, India
[3] Caring Hands Physiotherapy Ltd., Calgary, Canada
[4] Dr. Pradeep Sharma's Pain Management Clinic, New Delhi, Delhi, India
[5] Department of Rehabilitation Sciences, College of Applied Medical Sciences, King Saud University, Riyadh, Saudi Arabia

Corresponding author
Masood Khan,
raomasood22@gmail.com

## ABSTRACT

**Background:** Previous studies have examined the mode specificity of eccentric and concentric isokinetic training, but have reported conflicting results. Few studies have reported that eccentric and concentric isokinetic training are mode-specific, *i.e.*, they will increase only the eccentric or concentric strength, respectively. Other studies have reported that mode specificity does not exist. Therefore, this study aimed to assess the mode-specific effects of eccentric and concentric isokinetic training of the hamstring muscle at slow angular velocity on eccentric peak torque of the hamstring ($PT_{ecc}$), concentric peak torque of the quadriceps ($PT_{con}$), acceleration time of the hamstring ($AT_{hams}$) and quadriceps ($AT_{quad}$), deceleration time of the hamstring ($DT_{hams}$) and quadriceps ($DT_{quad}$), time to peak torque of the hamstring ($TPT_{hams}$) and quadriceps ($TPT_{quad}$), and functional Hamstring-to-Quadriceps ratio ($PT_{ecc}/PT_{con}$).

**Subjects:** A total of 30 participants were randomly divided into eccentric and concentric groups.

**Methods:** Two groups pre-test-post-test experimental design was used. In the eccentric and concentric groups, eccentric and concentric isokinetic training of hamstring muscle was performed respectively, at an angular velocity of $60°/s$ for 6 weeks duration. $PT_{ecc}$, $PT_{con}$, $AT_{hams}$, $AT_{quad}$, $DT_{hams}$, $DT_{quad}$, $TPT_{hams}$, $TPT_{quad}$, and $PT_{ecc}/PT_{con}$ were measured before and after the completion of training.

**Results:** In the eccentric group, a significant difference ($p < 0.05$) was observed in $PT_{ecc}$ (increased by 21.55%), $AT_{hams}$ (decreased by 42.33%), $AT_{quad}$ (decreased by 28.74%), and $PT_{ecc}/PT_{con}$ (increased by 17.59%). No significant difference ($p > 0.05$) was observed in $PT_{con}$, $TPT_{hams}$, $TPT_{quad}$, $DT_{hams}$, and $DT_{quad}$. In the concentric group, a significant difference ($p < 0.05$) was observed in $PT_{ecc}$ (increased by 12.95%), $AT_{hams}$ (decreased by 27.38%) $AT_{quad}$ (decreased by 22.08%), $DT_{quad}$ (decreased by 26.86%), and $PT_{ecc}/PT_{con}$ (increased by 8.35%). No significant difference ($p > 0.05$)

was observed in $PT_{quad}$, $TPT_{hams}$, TPTquad, and $DT_{hams}$. Between-group analysis revealed a significant difference ($p < 0.05$) only in $TPT_{quad}$; otherwise, in the rest of the parameters, no significant difference ($p > 0.05$) was observed.

**Conclusions:** Both eccentric and concentric isokinetic training of the hamstring for 6 weeks increased $PT_{ecc}$, $PT_{ecc}/PT_{con}$, and decreased $AT_{hams}$ and $AT_{quad}$. The effects of eccentric and concentric isokinetic training of the hamstring on $PT_{ecc}$, $PT_{ecc}/PT_{con}$, $AT_{hams}$, and $AT_{quad}$ were not mode specific.

## INTRODUCTION

Good muscular strength provides dynamic joint stabilization and helps prevent injury (*Holcomb et al., 2007*). Not only is the strength of individual muscles important, but the ratio of strength between agonist and antagonist muscles also play an important role in injury prevention (*Clanton & Coupe, 1998*; *Hewett et al., 1999*; *Hewett et al., 1996*; *Li et al., 1996*; *Orchard et al., 1997*). In the knee joint, the quadriceps and hamstring muscles act as agonist and antagonist muscles. The functional hamstring-to-quadriceps ratio is considered a more relevant and functional estimate of the muscular balance of the knee joint than the conventional concentric hamstring-to-concentric quadriceps ratio (*Aagaard et al., 1998*; *Ruas et al., 2015*). The functional hamstring-to-quadriceps ratio ($PT_{ecc}/PT_{con}$) is calculated as the ratio between the peak eccentric torque of the hamstring ($PT_{ecc}$) and peak concentric torque of the quadriceps ($PT_{con}$) because this pattern of activity is performed in these two muscles during a kick movement (*Ruas et al., 2019*).

Several studies in past have tried to examine the mode specificity of concentric and eccentric training of different muscles and at different angular velocities *i.e.*, whether concentric training will increase the concentric strength only or both concentric and eccentric strength and whether eccentric training will increase eccentric strength only or both eccentric and concentric strength.

A study by *Pavone & Moffat (1985)* concluded that eccentric and concentric training resulted in an equivalent increase in isometric strength in quadriceps muscle therefore according to them mode specificity of these training does not have a significant role in the quadriceps muscle. *Duncan et al. (1989)*, in their study, performed concentric and eccentric training of the quadriceps muscle for 6 weeks and reported that isokinetic concentric training at 120°/s resulted in a significant increase in concentric strength at 180°/s only and no significant increase in eccentric strength. They also stated that isokinetic eccentric training at 120°/s resulted in a significant increase in eccentric strength at all test speeds with minimal effect on concentric strength. Therefore, according to their study (*Duncan et al., 1989*), eccentric training of the quadriceps muscle is highly mode-specific, which means it will only cause a substantial increase in eccentric strength.

Therefore, there are conflicting reports on the mode specificity of eccentric and concentric strengthening exercises. Since concentric and eccentric strengths of agonist and

antagonist muscles are required during activities of daily living and sports, therefore, for optimal performance and injury prevention, individuals have to perform both types of exercise if mode specificity exists. If mode specificity does not exist, then one type of exercise may be sufficient to increase both concentric and eccentric strengths.

In addition to the peak torque, several other isokinetic variables also provide valuable information such as acceleration time (AT), deceleration time (DT), and time to peak torque (TPT). AT is defined as 'the time required by the muscle to accelerate to a preset dynamometer speed' (*Chen, Su & Chou, 1994*; *van Cingel et al., 2006*). DT is defined as 'the time required by the muscle to reach a zero speed from a preset dynamometer speed' (*Lobato et al., 2018*). TPT is defined as 'the time required by the muscle to reach the peak torque' (*Bračič et al., 2011*). These parameters are considered muscle recruitment parameters that indicate neuromuscular readiness of the muscle to produce maximum contraction (*Chen, Su & Chou, 1994*; *Miller et al., 2006*; *van Cingel et al., 2006*) and are considered essential for protection against injuries (*Maciel et al., 2020*).

Since both concentric and eccentric contractions of muscles are necessary for movements and an individual should have both types of sufficient strength for optimal functions therefore they have to perform both types of training if mode specificity exists. If mode specificity does not exist, then they do not need to perform both types of training. Since conflicting reports are available regarding the mode specificity of eccentric and concentric isokinetic training therefore one study was needed that can examine the mode specificity of these training in hamstring muscles at slow angular velocity. The present study aimed to examine the mode-specific effects of six weeks of isokinetic concentric and eccentric training of the hamstring muscle at slow angular velocity on the $PT_{ecc}/PT_{con}$, AT, DT, and TPT. We hypothesized that the effects of isokinetic concentric and eccentric training of the hamstring muscle at a slow angular velocity of 60°/s are mode-specific.

## MATERIALS AND METHODS

### Study design

A two-arm pre-test-post-test experimental design was used with random allocation of participants into the eccentric isokinetic training group and the concentration isokinetic training group.

### Participants

A minimum number of 30 participants is required for experimental research to make a valid generalization (*Fink, 2003*; *Kraemer & Blasey, 2015*), therefore, convenient sampling was performed and a total of 30 healthy male collegiate students between the ages of 18 and 28 were selected for the study (Table 1) (Fig. 1). These students were recreationally active and used to engage regularly in 1–5 h of physical activity per week. Participants who had already participated in lower extremity strength training and had pain in the hip, knee, ankle, or foot were excluded from the study. Furthermore, participants with knee joint injury, lower extremity deformity, any other lower extremity musculoskeletal disorder, or cardiorespiratory disease were excluded from the study. Participants involved in strength training of lower limbs were also excluded. Participants were randomly assigned to the
**Table 1 Demographic characteristics of participants, _n_ = 15 in each group, mean ± SD.**

|  | Eccentric group | Concentric group |
|---|---|---|
| Age (years) | 24.27 ± 1.48 | 24.93 ± 1.16 |
| Height (cm) | 172.67 ± 6.05 | 170.27 ± 5.23 |
| Weight (Kg) | 68.23 ± 9.97 | 65.31 ± 6.82 |
| BMI (Kg/m$^2$) | 22.83 ± 2.70 | 22.51 ± 2.03 |

**Note:**
BMI, Body mass index; SD, Standard Deviation.

eccentric isokinetic training group or the concentric isokinetic training group with 15 participants in each group by an examiner. For randomization, the lottery method and the http://www.randomization.com website were used. An expert physical therapist screened the participants and guided the participants throughout the training program. The participants and outcome assessor were unaware of the random allocation. This study was consistent with 'The Code of Ethics of the World Medical Association (Declaration of Helsinki)'. The ethics subcommittee, King Saud University, approved the study (file id: RRC-2021-11). The study was carried out in the university research laboratory and retrospectively registered in the Protocol Registration and Results System, https://clinicaltrials.gov/ (ID: NCT05229367). The protocol of the study can be found on the website https://www.protocols.io/ with https://dx.doi.org/10.17504/protocols.io.3byl4bx2zvo5/v1 (_protocols.io, 2022_). The risks and benefits of the study were discussed with each participant before the start of the intervention who voluntarily participated in the study and gave their signed informed consent.

## Outcome measures

- Eccentric peak torque of the hamstring (PT$_{ecc}$)
- Concentric peak torque of the quadriceps (PT$_{con}$)
- Functional hamstring-to-quadriceps ratio (PT$_{ecc}$:PT$_{con}$)
- Acceleration time of the hamstring (AT$_{hams}$)
- Acceleration time of the quadriceps (AT$_{quad}$)
- Deceleration time of the hamstring (DT$_{hams}$)
- Deceleration time of the quadriceps (DT$_{quad}$)
- Time to peak torque of the hamstring (TPT$_{hams}$)
- Time to peak torque of the quadriceps (TPT$_{quad}$)

## Instrumentation

- Biodex multijoint system isokinetic device (Biodex Multi-Joint System 4; Biodex Medical Inc., Shirley, NY, USA)
- Universal goniometer

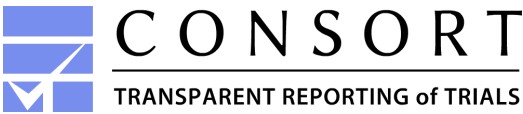

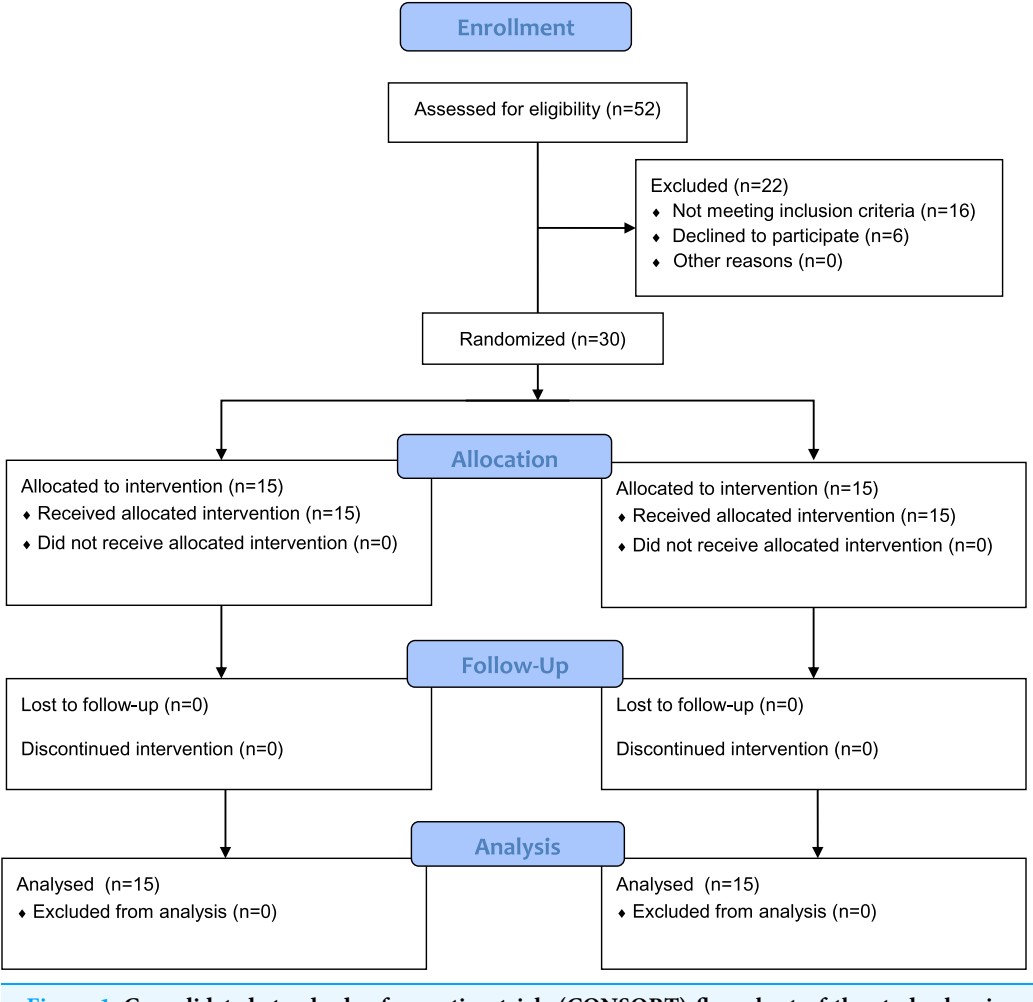

**Figure 1** Consolidated standards of reporting trials (CONSORT) flow chart of the study showing recruitment, allocation, and analysis of participants.

## Study protocol

The study was divided into three phases: A. Pre-intervention evaluation; B. Intervention; C. Post-intervention evaluation.

**A. Pre-intervention evaluation:** The non-dominant extremity was chosen for intervention. Before testing, the nondominant extremity was prepared by performing quadriceps, hamstring, and calf muscle stretching (30 s stretch, 30 s relax, and three repetitions). The participants were made to sit on an isokinetic dynamometer device. The axis of rotation of the isokinetic device was set parallel to the lateral femoral condyle. The range of motion at the knee joint was established between 5° knee

extension and 100° knee flexion. The baseline values of all outcome variables were recorded in both groups.

   i) *Measurement of $PT_{ecc}$ (hamstring)*: The isokinetic eccentric/concentric mode was selected in the dynamometer device. In this mode, the hamstring muscle first underwent eccentric contraction and then concentric contraction. During an eccentric contraction, angular velocity was set at 60°/s. Participants were asked to practice the movement three times with submaximal load to become familiar with the movement. Since only the eccentric peak toque measurement was intended, therefore, participants were asked to resist movement with maximal force during the eccentric phase and relax during the concentric phase. No peak torque measurement was made during the concentric phase. Participants were asked to perform the movement three times with maximum effort; then the mean of these three readings was taken as the baseline value. Baseline values of $AT_{hams}$, $DT_{hams}$, and $TPT_{hams}$ were also recorded.

   ii) *Measurement of $PT_{con}$ (quadriceps)*: After a gap of 10-min, the concentric peak torque of the quadriceps was measured. Concentric/concentric mode was selected on the dynamometer device. The angular velocity was selected at 60°/s for extension movement and 120°/s for flexion movement. Participants were asked to practice the movement three times with submaximal load to become familiar with the movement. Participants were asked to perform movement during the extension phase with maximum effort and relax during the flexion phase. A total of three readings were taken, then the mean of these three readings was taken as the baseline value. Baseline values of $AT_{quad}$, $DT_{quad}$, and $TPT_{quad}$ were also recorded.

**B. Intervention:** Two groups were created with an equal number of participants: the eccentric isokinetic training group and the concentric isokinetic training group. The preparation, warm-up, familiarization of participants, and isokinetic device arrangement were similar to the preintervention evaluation.

   i) *Eccentric isokinetic training of the hamstring muscle*: the hamstring muscle was trained eccentrically similarly as during the preintervention evaluation in the isokinetic dynamometer device at an angular velocity of 60°/s. The eccentric/concentric mode was selected. Participants were asked to resist movement during the eccentric phase with maximum force and relax during the concentric phase.

   ii) *Concentric isokinetic training of hamstring muscle*: hamstring muscle was trained concentrically at speed of 60°/s. Concentric/concentric mode was selected. Participants were asked to put maximal force during the flexion phase and relax during the extension phase.

   In both groups, a total of three sets with 10 repetitions in each set were performed with a 60 s rest period between two sets. The training was performed for 2 days per week for a total of 6-week duration. All participants were advised to refrain from other strength training of lower limbs like gym, sports, running, *etc*.

**Table 2 Dependent variables data, $n = 15$ each group, mean ± SD, and $p$-values for Shapiro-Wilk test of normality of baseline variables.**

| | Eccentric group | | | | Concentric group | | | |
|---|---|---|---|---|---|---|---|---|
| | Baseline | $p$-value | df | Post-intervention | Baseline | $p$-value | df | Post-intervention |
| $PT_{ecc}$ (N-m) | 119.77 ± 19.44 | 0.467 | 15 | 145.58 ± 31.60 | 115.34 ± 32.08 | 0.340 | 15 | 130.28 ± 27.15 |
| $PT_{con}$ (N-m) | 162.18 ± 21.42 | 0.349 | 15 | 166.73 ± 29.48 | 149.84 ± 35.12 | 0.292 | 15 | 157.65 ± 31.48 |
| $PT_{ecc}/PT_{con}$ | 0.74 ± 0.09 | 0.758 | 15 | 0.87 ± 0.08 | 0.76 ± 0.11 | 0.886 | 15 | 0.83 ± 0.13 |
| $TPT_{hams}$ (ms) | 979.33 ± 135.14 | 0.050 | 15 | 1,008.00 ± 145.95 | 1,037.33 ± 151.72 | 0.082 | 15 | 998.00 ± 224.60 |
| $TPT_{quad}$ (ms) | 447.33 ± 78.23 | 0.099 | 15 | 419.33 ± 85.06 | 494.67 ± 110.18 | 0.289 | 15 | 495.33 ± 98.11 |
| $AT_{hams}$ (ms) | 91.33 ± 42.74 | 0.145 | 15 | 52.67 ± 19.07 | 112.00 ± 48.28 | 0.588 | 15 | 81.33 ± 50.97 |
| $AT_{quad}$ (ms) | 53.33 ± 26.63 | 0.004* | 15 | 38.00 ± 11.46 | 57.33 ± 15.33 | 0.605 | 15 | 44.67 ± 13.02 |
| $DT_{hams}$ (ms) | 342.67 ± 64.30 | 0.234 | 15 | 370.00 ± 55.16 | 382.67 ± 53.24 | 0.500 | 15 | 375.33 ± 42.57 |
| $DT_{quad}$ (ms) | 826.00 ± 780.84 | 0.002* | 15 | 881.33 ± 683.86 | 940.67 ± 509.44 | 0.021* | 15 | 688.00 ± 295.71 |

Notes:
* Significant.
SD, Standard deviation; df, degree of freedom; $PT_{ecc}$, eccentric peak torque of hamstring; $PT_{con}$, concentric peak torque of quadriceps; $PT_{ecc}/PT_{con}$, ratio of eccentric peak torque of hamstring to concentric peak torque of quadriceps; $TPT_{hams}$, Time to peak torque of hamstring muscle; $TPT_{quad}$, Time to peak torque of quadriceps muscle; $AT_{hams}$, acceleration time of hamstring; $AT_{quad}$, acceleration time of quadriceps; $DT_{hams}$, deceleration time of hamstring; $DT_{quad}$, deceleration time of quadriceps; N-m, Newton-meter; ms, milliseconds.

**C. Post-intervention evaluation:** Post-intervention evaluation was performed 48 h after completion of the training. Following the similar method described in the pre-intervention evaluation, all outcome variables were measured in both groups.

## Data analysis

SPSS software, version 26 (SPSS Inc., Chicago, IL, USA), was used for data analysis. The normal distribution of baseline values of dependent variables was assessed using the Shapiro-Wilk test of normality. All baseline values of dependent variables showed normal distribution except for $AT_{quad}$ and $DT_{quad}$ in the eccentric group and $DT_{quad}$ in the concentric group, therefore for all the dependent variables, non-parametric tests were used. For with-in group and between-group analysis, the Wilcoxon signed-rank test and the Mann–Whitney U test were performed, respectively. Cohen's d was used to calculate the effect size in between-group analysis. The following categories of Cohen's d were considered: d = 0.2 as small effect size, d = 0.5 as medium effect size, and d = 0.8 as large effect size. The confidence interval was set at 95% and the results were considered significant with a $p$-value < 0.05.

## RESULTS

A total of 30 participants' data was statistically analyzed. Table 2 contains data for dependent variables and normality test results. Figures 2 and 3 depict the comparison of baseline and post-intervention values in both groups.

### Within group analysis (Wilcoxon signed-rank test)

Table 3 contains within group results for both groups.
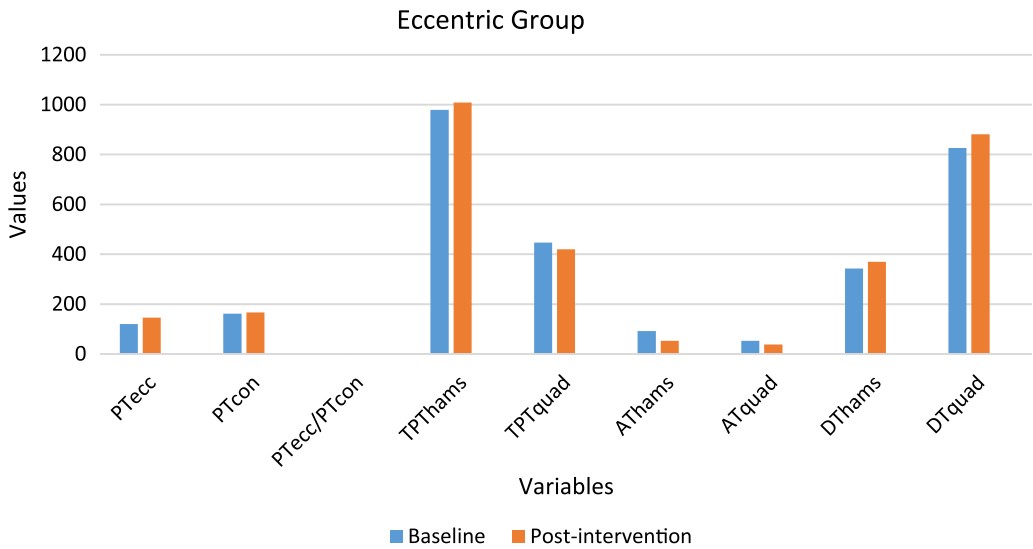

**Figure 2 Graph depicting the variables baseline and post-intervention values in the eccentric group.**

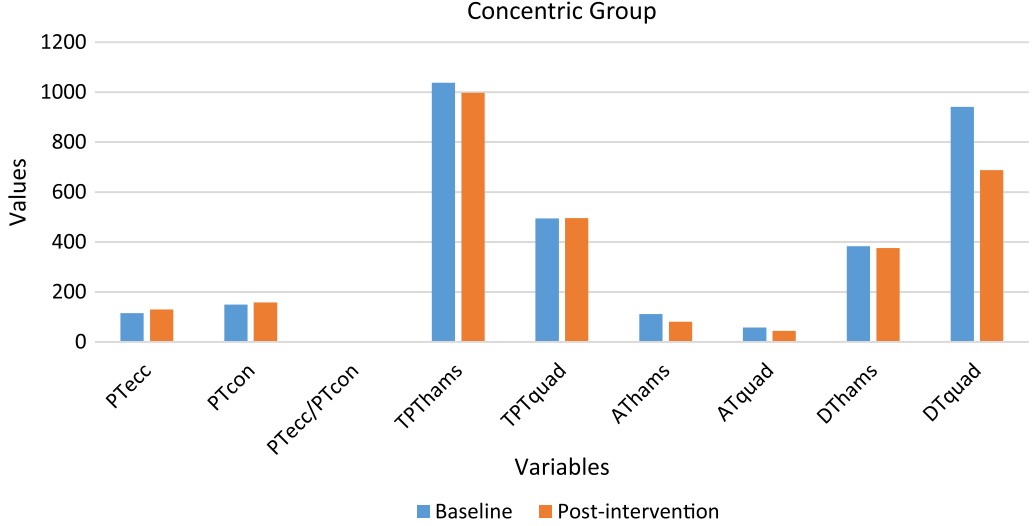

**Figure 3 Graph depicting the variables baseline and post-intervention values in the concentric group.**

## Eccentric group

A significant difference ($p < 0.05$) was observed in $PT_{ecc}$ (increased by 21.55%), $AT_{hams}$ (decreased by 42.33%), $AT_{quad}$ (decreased by 28.74%), and the $PT_{ecc}/PT_{con}$ ratio (increased by 17.59%). No significant difference ($p > 0.05$) was observed in $PT_{con}$, $TPT_{hams}$, $TPT_{quad}$, $DT_{hams}$, and $DT_{quad}$.

## Concentric group

A significant difference ($p < 0.05$) was observed in $PT_{ecc}$ (increased by 12.95%), $AT_{hams}$ (decreased by 27.38%), $AT_{quad}$ (decreased by 22.08%), $DT_{quad}$ (decreased by 26.86%), and

**Table 3 Within group (Wilcoxon signed-rank test) results for both groups, mean difference ± SD, *p*- and Z-values.**

| | Eccentric group | | | Concentric group | | |
|---|---|---|---|---|---|---|
| | Mean difference ± SD | *p*-value | Z | Mean difference ± SD | *p*-value | Z |
| $PT_{ecc}$ Post − $PT_{ecc}$ Baseline | 25.81 ± 22.34 | 0.002* | −3.107 | 14.94 ± 12.47 | 0.001* | −3.233 |
| $PT_{con}$ Post − $PT_{con}$ Baseline | 4.55 ± 18.60 | 0.730 | −0.345 | 7.81 ± 15.96 | 0.124 | −1.538 |
| $TPT_{hams}$ Post − $TPT_{hams}$ Baseline | 28.66 ± 174.43 | 0.396 | −0.848 | −39.33 ± 192.51 | 0.730 | −0.346 |
| $TPT_{quad}$ Post − $TPT_{quad}$ Baseline | −28.00 ± 115.02 | 0.232 | −1.195 | 0.66 ± 71.36 | 0.894 | −0.133 |
| $AT_{hams}$ Post − $AT_{hams}$ Baseline | −38.66 ± 43.07 | 0.004* | −2.868 | −30.66 ± 44.31 | 0.031* | −2.162 |
| $DT_{hams}$ Post − $DT_{hams}$ Baseline | 27.33 ± 84.47 | 0.271 | −1.100 | −7.33 ± 63.41 | 0.700 | −0.385 |
| $AT_{quad}$ Post − $AT_{quad}$ Baseline | −15.33 ± 25.03 | 0.026* | −2.222 | −12.66 ± 15.79 | 0.009* | −2.630 |
| $DT_{quad}$ Post − $DT_{quad}$ Baseline | 55.33 ± 394.07 | 0.507 | −0.664 | −252.66 ± 450.75 | 0.026* | −2.229 |
| $PT_{ecc}/PT_{con}$ Post − $PT_{ecc}/PT_{con}$ Baseline | 0.13 ± 0.09 | 0.002* | −3.170 | 0.0642 ± 0.09 | 0.026* | −2.229 |

Notes:
* Significant.
SD, Standard deviation; $PT_{ecc}$, eccentric peak torque of hamstring; $PT_{con}$, concentric peak torque of quadriceps; $TPT_{hams}$, Time to peak torque of hamstring muscle; $TPT_{quad}$, Time to peak torque of quadriceps muscle; $AT_{hams}$, acceleration time of hamstring; $DT_{hams}$, deceleration time of hamstring; $AT_{quad}$, acceleration time of quadriceps; $DT_{quad}$, deceleration time of quadriceps. $PT_{ecc}/PT_{con}$, ratio of eccentric peak torque of hamstring to concentric peak torque of quadriceps.

the $PT_{ecc}/PT_{con}$ ratio (increased by 8.35%). No significant difference ($p > 0.05$) was observed in $PT_{quad}$, $TPT_{hams}$, $TPT_{quad}$ and $DT_{hams}$.

## Between-group analysis (Mann-Whitney test)

Table 4 contains between-group results.

A significant difference ($p < 0.05$) was observed only in $TPT_{quad}$; otherwise, in all other parameters ($PT_{hams}$, $PT_{quad}$, $TPT_{hams}$, $AT_{hams}$, $AT_{quad}$, $DT_{hams}$, $DT_{quad}$, and $PT_{ecc}/PT_{con}$) no significant difference ($p > 0.05$) was observed between both groups.

## DISCUSSION

This study aimed to examine the effects of 6-weeks eccentric and concentric isokinetic training of hamstring muscle on $PT_{ecc}$, $PT_{con}$, $AT_{hams}$, $AT_{quad}$, $DT_{hams}$, $DT_{quad}$, $TPT_{hams}$, $TPT_{quad}$, and $PT_{ecc}/PT_{con}$. In the present study, eccentric isokinetic training of the hamstring muscle of 6-week duration increased the $PT_{ecc}$, $PT_{ecc}/PT_{con}$, and decreased the $AT_{hams}$, $AT_{quad}$. Concentric isokinetic training of the hamstring muscle also increased $PT_{ecc}$, $PT_{ecc}/PT_{con}$, decreased $AT_{hams}$, $AT_{quad}$, and $DT_{quad}$. When eccentric isokinetic training was compared with concentric isokinetic training then no significant difference was observed between them except for $TPT_{quad}$, therefore eccentric and concentric isokinetic training were equally effective in improving $PT_{ecc}$, $PT_{ecc}/PT_{con}$, reducing the $AT_{hams}$ and $AT_{quad}$. Therefore, according to the findings of this study, the effects of eccentric and concentric isokinetic training of hamstring muscle on $PT_{ecc}$, AT and $PT_{ecc}/PT_{con}$ are not mode-specific. The findings of the present study are supported and not supported by previous studies. Several previous studies reported the mode specificity of both concentric and eccentric strength training, contrary to the findings of the present study (*Higbie et al., 1996*; *Hortobagyi et al., 1996*; *Seger et al., 1998*; *Tomberlin et al., 1991*). A study by *Mjølsnes et al. (2004)* reported a significant increase in the functional hamstring-to-quadriceps ratio after 10 weeks of eccentric training of hamstring muscle;

**Table 4 Between-group (Mann-Whitney U Test) results, Z-, p-, and Cohen's d values.**

|  | Z | *p*-value | Cohen's d | *Post hoc* power | df |
|---|---|---|---|---|---|
| $PT_{ecc}$ Post (N-m) | −1.141 | 0.254 | 0.51 | 0.26 | 26.64 |
| $PT_{con}$ Post (N-m) | −0.726 | 0.468 | 0.29 | 0.11 | 26.64 |
| $TPT_{hams}$ Post (ms) | −0.415 | 0.678 | 0.05 | 0.05 | 26.64 |
| $TPT_{quad}$ Post (ms) | −2.433 | 0.015* | 0.82 | 0.56 | 26.64 |
| $AT_{hams}$ Post (ms) | −1.722 | 0.085 | 0.74 | 0.47 | 26.64 |
| $DT_{hams}$ Post (ms) | −0.646 | 0.518 | 0.10 | 0.05 | 26.64 |
| $AT_{quad}$ Post (ms) | −1.536 | 0.125 | 0.54 | 0.28 | 26.64 |
| $DT_{quad}$ Post (ms) | −0.062 | 0.95 | 0.36 | 0.15 | 26.64 |
| $PT_{ecc}/PT_{con}$ Post | −1.182 | 0.237 | 0.35 | 0.14 | 26.64 |

**Notes:**
* Significant.
$PT_{ecc}$, eccentric peak torque of hamstring; $PT_{con}$, concentric peak torque of quadriceps; $TPT_{hams}$, Time to peak torque of hamstring muscle; $TPT_{quad}$, Time to peak torque of quadriceps muscle; $AT_{hams}$, acceleration time of hamstring; $DT_{hams}$, deceleration time of hamstring; $AT_{quad}$, acceleration time of quadriceps; $DT_{quad}$, deceleration time of quadriceps; $PT_{ecc}/PT_{con}$, ratio of eccentric peak torque of hamstring to concentric peak torque of quadriceps; N-m, Newton-meter; ms, milliseconds; df, degree of freedom.

however, concentric training did not have any effects. In the present study, participants performed isokinetic concentric exercises in an isokinetic dynamometer device, however, *Mjølsnes et al. (2004)* used isotonic hamstring curls in a traditional hamstring curls machine. *Duncan et al. (1989)* in their study on the mode specificity of concentric and eccentric exercise training of the quadriceps muscle reported that the eccentric mode of isokinetic exercises had highly specific effects on muscle strength compared to the concentric mode, which means that eccentric exercise will increase eccentric force only. *Ruas et al. (2018)* examined the effects of three types of muscle action training protocols of 6-weeks duration, on several parameters including functional hamstring to quadriceps ratio. The following three training protocols were used in their study: concentric quadriceps and eccentric hamstring, eccentric quadriceps and eccentric hamstring, and concentric quadriceps and concentric hamstring. The eccentric quadriceps and eccentric hamstring groups showed significant increases in functional hamstring to quadriceps ratio and eccentric peak torque. For concentric peak torque, there was no difference between the groups (*Ruas et al., 2018*).

*Ryan, Magidow & Duncan (1991)* reported findings similar to the present study and concluded that the effects of eccentric isokinetic training of hamstring muscle at 120°/s are not mode-specific and resulted in an increase in both eccentric and concentric strength at certain angular speeds. Other previous eccentric training studies on hamstring muscle also reported that eccentric isokinetic training is not mode-specific (*Fridén et al., 1983*; *Komi & Buskirk, 1972*).

Another finding in the present study was a reduction in $AT_{hams}$ and $AT_{quad}$ after 6 weeks of eccentric and concentric isokinetic training of the hamstring muscle. Concentric isokinetic training also reduced $DT_{quad}$. Shorter AT and DT may play a role in injury prevention. It is suggested that during abrupt movements, the muscles around the joint should contract rapidly to stabilize it and prevent injuries; if this neuromuscular

recruitment is delayed, then the joint may be more prone to injury (*van Cingel et al., 2006*). Other authors have suggested that the shortest muscle reaction time plays a significant role in injury prevention, especially in the ankle joints (*van Cingel et al., 2006*), elbow joints (*Scattone-Silva et al., 2012*), and knee joints (*Johnson, Palmieri-Smith & Lepley, 2018*) joints. Since very few studies have been performed on the effects of isokinetic training on AT and DT, it is difficult to compare the findings of this study with the findings of similar previous studies.

Since both eccentric and concentric isokinetic training of hamstring muscle were effective in increasing the $PT_{ecc}$, $PT_{ecc}/PT_{con}$ and decreasing the $AT_{hams}$ and $AT_{quad}$, therefore either of these two training can be performed by athletes to achieve the desired results in these parameters.

## CONCLUSIONS

Both eccentric and concentric isokinetic training of the hamstring muscle for 6 weeks was effective in increasing $PT_{ecc}$, $PT_{ecc}/PT_{con}$, and decreasing $AT_{hams}$ and $AT_{quad}$. The effects of eccentric and concentric isokinetic training of the hamstring muscle on $PT_{ecc}$, $PT_{ecc}/PT_{con}$, $AT_{hams}$, and $AT_{quad}$ were not mode-specific, that is, concentric isokinetic training of the hamstring muscle also increased $PT_{ecc}$.

### Limitations and future research

The present study has several limitations also. Only male participants were recruited due to which the results of the present study cannot be generalized to the female population. The present study included recreationally active participants, not professional athletes. Professional athletes may undergo neuromuscular adaptations due to their long training career; therefore, we may see different responses to the isokinetic training used in the present study in professional athletes. In the present study, the mode specificity of eccentric and concentric isokinetic training of a lower limb muscle (hamstring) was examined; however, in sports, the agonist-antagonist relationship is important in the upper limbs also; therefore, future research should examine the mode specificity in upper limb muscle also, *e.g.*, triceps brachii. In the present study, peak torque was measured at only one angular velocity, *i.e.*, 60°/s; the same response may not be observed at faster angular velocities; therefore, future research should examine the mode-specificity at faster angular velocities (120°/s or 180°/s). Future research should also examine EMG activity of the muscles along with isokinetic parameters so that neuromuscular activity can be further studied.

### Funding

This research is funded by the Researchers Supporting Project number (RSP-2021/382), King Saud University, Riyadh, Saudi Arabia. The funders had no role in study design, data collection and analysis, decision to publish, or preparation of the manuscript.

## Grant Disclosures
The following grant information was disclosed by the authors:
King Saud University, Riyadh, Saudi Arabia: RSP-2021/382.

## Competing Interests
Kushla Nand Sharma, Unaise Abdul Hameed and Anita Kumari are employed by Sadhya Physiotherapy Clinic, Caring Hands Physiotherapy Ltd. and Dr. Pradeep Sharma's Pain Management Clinic respectively.

## Author Contributions
- Kushla Nand Sharma conceived and designed the experiments, performed the experiments, analyzed the data, prepared figures and/or tables, and approved the final draft.
- Nishat Quddus conceived and designed the experiments, analyzed the data, prepared figures and/or tables, and approved the final draft.
- Unaise Abdul Hameed conceived and designed the experiments, analyzed the data, authored or reviewed drafts of the article, and approved the final draft.
- Sohrab Ahmad Khan conceived and designed the experiments, prepared figures and/or tables, and approved the final draft.
- Anita Kumari performed the experiments, authored or reviewed drafts of the article, and approved the final draft.
- Ahmad H. Alghadir performed the experiments, authored or reviewed drafts of the article, and approved the final draft.
- Masood Khan performed the experiments, prepared figures and/or tables, authored or reviewed drafts of the article, and approved the final draft.

## Human Ethics
The following information was supplied relating to ethical approvals (*i.e.*, approving body and any reference numbers):
The ethics subcommittee, King Saud University, approved the study (file id: RRC-2021-11).

## Data Availability
The baseline and post-intervention measurements of all participants in both groups are available in the Supplemental File.

## Clinical Trial Registration
The following information was supplied regarding Clinical Trial registration:
NCT05229367.

## Supplemental Information
Supplemental information for this article can be found online at http://dx.doi.org/10.7717/peerj.13842#supplemental-information.

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
