# Peer review of "Mode-specific effects of concentric and eccentric isokinetic training of the hamstring muscle at slow angular velocity on the functional hamstrings-to-quadriceps ratio-a randomized trial"

_PeerJ, doi:10.7717/peerj.13842_

## Round 0.1 · original submission · Major Revisions

Please pay close attention to the comments and concerns by each reviewer and highlight your responses clearly.

Reviewer 1 ·

Basic reporting

Clear language is used throughout, though a few minor grammatical and syntactical errors could be addressed, I found it to be an enjoyable read. The data is reported unambiguously, but I would note the absence of units for the torque values as well as time. My assumption would be that these are in Newtons and Milliseconds, but I ask that you input them where appropriate. All references were cited appropriately, and the statistics reported appropriately

Experimental design

This study employed valid measures to assess the functional hamstring to quadricep ratios and other intervention variables of note. The two-armed, pre/post, RCT design was a well utilized approach to address the research questions. Appropriate statistical tests were used and reported.

Validity of the findings

These findings support previous studies wherein other authors have found that the effects of eccentric and concentric isokinetic training of the hamstring muscles are not mode specific, while taking care to mention that conflicting data also exists. The limitations of the present study are well addressed, and overall these data support the conclusions made by the authors.

Additional comments

Overall a parsimonious study design with appropriate statistical reporting. All supplemental information appears to be in line with PeerJ guidelines, including tables, figures, data reporting, and consort flow chart.

Reviewer 2 ·

Basic reporting

General comments

Thank you for the opportunity to read and comment on the manuscript entitled “Mode-specific effects of concentric and eccentric isokinetic training of the hamstring muscle at slow angular velocity on the functional hamstring-to-quadriceps ratio – a randomized trial”.

The present study shows that 6 weeks of knee flexor eccentric or concentric training increases concentric and eccentric peak torques, functional ratio, and other mechanical variables retrieved from the isokinetic dynamometer. Surprisingly, the changes in most of the dependent variables were dependent on the type of training performed, suggesting no contraction mode dependency. While these results are interesting, surprisingly, it violates the principle of training specificity. Overall, the manuscript needs improvement in its structure and, particularly, writing. I would suggest the authors check for more scientific language.

The manuscript, in some ways, lacks clarity as to why it needs to be performed. It is not immediately clear why the research is important and what it adds to our current body of knowledge of contraction mode adaptations. For example, in the introduction, the authors keep mentioning there’s contradictory findings regarding the effect of contraction mode effects on concentric and eccentric forces or H:Q functional ratios. The authors only cite 2 studies from the 80’s to show this contraction – there is, clearly, many more studies in the field that investigated the effects of contraction mode specificity on strength adaptations and H:Q ratio, i.e., individual studies showing the effect of eccentric training on eccentric and concentric strength and vice-versa. It is also unclear why it is important to test the effects of concentric and eccentric training on H:Q functional ratio and AT, DT, TPT. What is the practical and functional implication for this? Given the lack of information to justify the study, it was not surprising that no hypotheses were provided. This needs to be assessed. If the above is added, it will form a much stronger manuscript.

The methods need further clarification: on what bases were 30 participants were “required for experimental research to make a valid generalization”? If you’re not performing a-priori power calculation then you should, at least, provide post-hoc statistical power. This can be included in the statistics along with more details, e.g., 95% CI.
Finally, I believe the discussion needs further details. The manuscript lacks more recent studies (e.g., Gioftsidou et al 2006 (10.2466/pms.103.1.151-159); Ruas et al 2018 and 2019 - 10.1519/JSC.0000000000002134 &10.3390/sports7100221).

I believe the manuscript requires some improvements. I wish the authors all the best of luck for the new changes.

Experimental design

NA

Validity of the findings

NA

---

## Round 0.2 · accepted · Accept

Congratulations on your acceptance!